# The Experiences of Psychiatric Mental Health Nurse Practitioners with Clinical Supervision in South Korea: A Grounded Theory Approach

**DOI:** 10.3390/ijerph192315904

**Published:** 2022-11-29

**Authors:** Sung-Nam Lee, Hyun-Jin Kim

**Affiliations:** 1Nursing Department, Seoul Metropolitan Eunpyeong Hospital, Seoul 03476, Republic of Korea; 2School of Nursing, Hanyang University, Seoul 04763, Republic of Korea

**Keywords:** psychiatric nursing, clinical supervision, grounded theory

## Abstract

Clinical supervision (CS) helps improve expertise and job satisfaction in nursing staff, but its grounded research is limited. This study was conducted to derive a grounded theory based on the lived experiences of psychiatric mental health nurse practitioners (PMHNPs) in clinical supervision. Data were collected from January to April 2018 through in-depth unstructured interviews with 19 PMHMPs. Supervision of mental health nurses was necessary because of the “lack of ability to integrate theory and practice” and “difficulty working alone”. The “poor supervision system” has been strengthened. The nurses used strategies such as “asking for help”, “intensive training and sharing with the supervisor”, “modeling of the supervisor and developing competencies”, “continuing self-reflection and learning”, and “participating in professional activities”, according to the level of “personality characteristics”, “institutional supervision policy”, and “relationship with the supervisor”. Consequently, the core objective of “supporting each other and becoming healthcare experts” was achieved. These findings can be used as a basis for education, practices, research, and policy development of mental health nursing. This study highlights areas for policy improvement to ensure that high-quality mental health nursing can be achieved through appropriately targeted CS.

## 1. Introduction

Psychiatric mental health nurse practitioners (PMHNPs) are mental health specialists (MHSs) according to the Korean 1995 Promotion of Mental Health and Welfare Service Support for Mental Illness Act. MHSs include PMHNPs, mental health clinical psychologists, and mental health social workers, who are categorized as Level 1 and Level 2 specialists. Level 2 specialists must undergo 150 h of theoretical education and 850 h of practice per year under clinical supervision (CS), whereas Level 1 specialists must be trained for 3 years [1]. 

According to the World Health Organization (WHO), in South Korea, in 2014, the number of mental health professionals per 1,00,000 population was 14.7, which was less than half of the 31.9 reported in the United Kingdom and Australia, which are two similar high-income countries [2]. Furthermore, the number of trained PMHNPs was reported to have decreased by 47% per year in the 20 years before 2020 [3]. In 2019, the proportion of full-time PMHNPs working in local community mental health facilities in South Korea was 27% [4]. 

Currently, the average work experience of full-time PMHNPs at mental health welfare centers in South Korea is 3.3 years [3]. The quality of mental health services is of great concern, as there are few PMHNPs with advanced skills working in mental health welfare centers, which renders providing effective peer supervision virtually impossible. However, the Nursing University Association in Seoul stopped training PMHNPs in 2021, and an advanced Level 1 training program has never been established. 

Patients with severe mental illnesses have two to three times higher prevalence of a physical disease than do mentally healthy individuals [5]. Thus, the lack of PMHNPs has serious implications, such as less effective multidisciplinary teams [3], decreased quality of care, and a greater likelihood of overlooking an increased risk of suicide [6]. A survey in South Korea found that 80.6% of MHSs have experienced various forms of verbal, physical, and sexual assault at work, alongside having to deal with suicide attempts of patients [3]. Such events tend to induce further exhaustion and job turnover among MHSs [7].

In medicine, psychology, social work, and nursing, CS is widely used as an effective method for professional training and education [8]. It is useful for teaching psychiatric concepts and therapeutic principles of human relationships [9] and in integrating theory and practice in these areas [10]. Continuous CS participation affects the employment and maintenance of PMHNP employees, as well as quality services [11].

In studies conducted outside South Korea, systematic CS has been shown to improve the quality of care [12] by reducing patient mortality and complications [13] while increasing autonomy and preventing burnout [14]. In particular, in mental health nursing, CS helps improve the quality of medical care by reducing burnout and stress, as well as improving professionalism and job satisfaction [6,15,16,17,18,19]. CS is widely used internationally, with research on it steadily increasing [11,20]. However, there is still a need to determine an agreed definition of CS, and CS practices in nursing lack both rigorous research assessment and empirical evidence in relation to meaningful outcomes [20]. In addition, it is difficult to find grounded theoretical studies on the development of CS experience in PMHNPs of long-term employment.

As of 2022, there are only two studies related to CS experienced by PMHNPs in South Korea: “phenomenological research” [21] and “conceptual analysis” [22]. In a previous research study in South Korea, no evidence of CS education or use of CS terminology was found in South Korean nursing schools. An unclear understanding of CS hinders CS research and its practical development [23] in South Korea. Therefore, considering this situation and the limitations of previous studies in South Korea, research that comprehensively explores the CS experience of mental health nurses is needed. This study aimed to explore and analyze the CS experiences of mental health nurses to determine key components in relation to the experience and facilitate a more comprehensive and evidence-based understanding of CS.

## 2. Materials and Methods

### 2.1. Aims and Research Design

This qualitative study used grounded theory to comprehensively explore the CS experience of PMHNPs in the sociocultural context of South Korea. Grounded theory is a qualitative research methodology, based on the concept of symbolic reciprocity, that can be used to advance conceptual understanding in fields where appropriate understanding is lacking or to clarify existing theories [24]. Grounded theory derives insight from collected data in relation to real-life phenomena and is likely to provide a more comprehensive understanding of CS experiences [24]. 

Thus, this study posed the following research question: *“How have PMHNPs experienced CS?”* The results of this study can be used to inform the development of clinical mental health practice and mental health nursing curriculum in South Korea.

### 2.2. Research Participants

Nineteen participants currently working as PMHNPs were included in this study. Recruitment of participants was objective-based snowball sampling, in which the first participant recommended the next participant. The inclusion criteria were as follows: PMHNPs who (1) had received supervision for more than one year, (2) had more than 5 years of working experience related to mental health, and (3) worked in mental health hospitals or local community mental health facilities. Data collection was completed after confirming that theoretical saturation was reached when no new concepts emerged.

The 19 participants were women aged between 40 and 65 years, with a mean work experience of 20 years in mental health care. There were not any male participants in the study, because 98.4% of PMHNPs in South Korea are female [25]. At the time of the interviews, nine participants worked in hospitals, ten worked in local communities, and fifteen had experience working in both. The participants were clinically supervised by more than three supervisors on average, and the mean period under CS was 10 years. There were 15 Level 1 PMHNPs and 4 Level 2 PMHNPs (Table 1).

### 2.3. Ethical Considerations

This study was approved by the Institutional Review Board of Hanyang University (HYI-18-011-1). The participants were informed about the purpose and method of the study, confidentiality, anonymity, recording of the interviews, and the possibility of publication. Written consent was obtained from all participants. The participants were also informed that they could withdraw from the study at any time without any penalties. During the transcription process, all personal information was encoded, using symbols, and secured so that only the researchers could interpret it, in accordance with the relevant laws.

### 2.4. Data Collection

Data were collected between January and April 2018. The interviews lasted 70 min on average per person, but they could be extended if further information was needed during the analysis. The interviews were conducted directly by the researcher at a time and place desired by the participants. 

The in-depth interviews initially focused on the participants’ everyday life experiences to help them relax. The interview started with an open-ended question, “Tell us about your CS experience”, to prompt the participants to consider their CS during their training, after which follow-up questions were posed to explore and assess the participants’ experiences during their work as PMHNPs, such as the following: “Have you ever felt the need for CS? Please explain the situation”, “Have your thoughts or behaviors changed after CS? Please provide specific details”, “What difficulties did you experience during your CS?”, “If you had any difficulties, how did you overcome them?”, “What did you want to achieve with CS?” and “What do you think is necessary to further develop CS?” 

All interviews were audio-recorded and transcribed verbatim into Korean. The facial expressions, tones, and behaviors of the participants were carefully observed, and non-verbal responses were noted. The interviews were terminated once no new meaningful data could be observed and when the categories and dimensions reached theoretical saturation. 

### 2.5. Data Analysis and Methodological Rigor

Data were collected according to Strauss and Corbin’s [25] grounded theory and analyzed at the same time in open coding, axial coding, and selective coding stages. Researchers continued their comparative analysis of the original data with theoretical sensibility, grouping each concept and developing it more abstractly into categories. In this process, if there was a difference of opinion in the analysis, the opinions were integrated through the process of reconfirming the original data. The continuous comparative analysis method has been found to be useful in inductively identifying central phenomena, core categories, and strategies among individuals in workplaces [25].

In the present study, data collection and analysis reached theoretical saturation at the 19th participant, with no new themes and patterns emerging in the data. For final validation, the results for each stage were reviewed two to three times by three researchers who had studied or published articles on grounded theory or qualitative research methodology, ten participants, and five Level 1 PMHNPs who were working as PMHNPs but did not participate in the study.

## 3. Results

In open coding, a total of 97 concepts with 14 categories and 35 subcategories were derived (Table 2). A paradigm model connecting the relationships between each category is presented in Figure 1. The process of CS experience of PMHNPs was divided into four stages: “asking for help”, “intensive training and accompanying”, “developing competency”, and “autonomous cooperation”. The main category explaining the process of CS experience of PMHNPs was “supporting each other and becoming a healthcare expert”. 

The frequency of the analyzed subcategories was added to Table 2 by classifying the verbatim (A4 Korean page 287, Excel raw data analyzed by semantic units).

### 3.1. Causal Conditions: “Difficulty Working Alone”

In this study, the causal condition was “difficulty working alone”, and the subcategories were “poor working conditions” and “being fearful and tense”. Participants first started as general nurses and were asked to become mental health nurses without any practical training, due to a shortage of manpower. They experienced low self-esteem, faced ethical conflicts, and considered changing jobs, as they functioned in a poor working environment.


*“There are many patients in the psychiatric ward, but one nurse is tasked with doing the work of three people. There are many ethical conflicts and I wanted to quit every day”.*
(Participant D)


*“The patients fight and throw things at each other. The staff get physically hit and sometimes even strangled from behind by the patients… We are just asked to understand the patients without any treatment or apologies. It was hard and frustrating”.*
(Participant D)

### 3.2. Contextual Condition: “Poor Clinical Supervision System”

The contextual condition affecting the central phenomenon was “poor clinical supervision system”, and the subcategories were “superficial clinical supervision” and “desire for systematic practical clinical supervision”. CS is a fundamental process for the training of PMHNPs. However, CS for experienced workers after training was lacking. As CS is mostly superficial, focused on tasks, the participants responded that they wanted a variety of continuous CS. The participants had their Level 2 certificate as specialists and obtained clinical work experience in the ward. Thus, they were confident in case management; however, they lacked experience with administrative work, and a poor CS system led to limited CS. The participants had received superficial CS during their PMHNP training courses; however, after their training courses, there was a “*poor supervision system*”; therefore, the PMHNPs expressed a strong preference for systematic and practical CS.


*“We were not clinically supervised for tasks other than case management during the training. I really wanted to gain various CS experiences. Once the training is completed, you are no longer clinically supervised. You have nobody to direct your questions to, and it’s frustrating”.*
(Participant K)


*“In the local community mental health welfare centers, there are only a few nurses. The team leaders were also social workers; hence, it was difficult to be clinically supervised”.*
(Level 1 PMHNP)


*If there had been at least one clinical supervisor… (I would not have changed jobs)…”*
(Participant N)

### 3.3. Central Phenomenon: “Lack of Ability to Integrate Theory and Practice”

The central phenomenon in this study was the “lack of ability to integrate theory and practice”. The subcategories were “feeling of lack of competence”, “feeling lethargic”, “feeling burdened”, “feeling exhausted”, and “difficulty adapting to the diversity of mental health sites”, leading to the feeling of “lack of ability to integrate theory and practice”. 

The participants could not understand the requests or behaviors of the clients and felt confused and frustrated as they did not know how to respond. They lost confidence and felt “lack of competence”.


*“Mental health nursing requires empathy, as we talk to clients. However, there were many times when I felt incompetent and ignorant, not knowing what to do because of my lack of counseling skills”.*
(Participant H)

The participants tried to apply what they had learned during training; however, they repeatedly made mistakes and felt “lethargic” and “daunted”.


*“I have worked in psychiatry wards for a long time, and I feel lethargic, watching most of the patients become chronic patients. I feel like I am also developing a slight psychotic state, and I started to worry”.*
(Participant L)

The participants observed that customized holistic mental health nursing care for patients, who were not generally aware of their conditions, was a “burden”, unlike physical health nursing.


*“The client believes that the family will all die because of him and is highly anxious. I gave him medications and talked to him through counselling sessions, but the delusions did not go away easily”.*
(Participant F)


*“We have to be professionals on the front line, helping clients to live well. However, they always have different dynamics and situations; hence, psychiatric nursing is a burden”.*
(Participant B)

Participants became depressed as they listened to and empathized with the stories of the patients. They suffered vicarious trauma due to psychological shock and became “exhausted”, leading to thoughts of changing jobs.


*“When we hear shocking stories from the clients, we keep thinking about what we should tell the client the next day even after work. These situations are depressing and make us feel terrified. We think about quitting all the time”.*
(Participant Q)


*“Psychiatric nursing is extremely exhausting. It is often difficult to educate clients. You cannot give them positive energy by just thinking about your work, and sometimes you simply cannot change the clients”.*
(Participant R)

In hospitals, many tasks are doctor-ordered. In contrast, in community mental health sites, there is a lot of independent work. The participants had “difficulties in adapting” and left community mental health nursing jobs, as they were unfamiliar with performing bigger tasks such as reviewing/establishing mental health policies and planning the lives of clients.


*“In hospitals, clients are discharged as their symptoms improve. However, in local community centers, taking a careful look at the lives of patients with mental illness requires different perspectives. This was fairly difficult at first”.*
(Participant C)

### 3.4. Intervening Condition

Intervening conditions that promoted action/interaction strategies were “personality characteristics”, “institutional clinical supervision policy”, and “relationship with clinical supervisors”.

As the participants voluntarily chose to work in psychiatric nursing, “personal characteristics” of actively striving for professional improvement and accepting CS with a positive mind helped them become better PMHNPs.


*“It was embarrassing to see my clinical supervisor, but I was honest about my wrongdoing and mistakes and tried my best to learn”.*
(Participant A)


*“I wanted to learn, grow, and become a better nurse rather than rigidly performing the given duties”.*
(Participant C)

Participants continued or stopped CS according to the “institutional clinical supervision policy”.


*“There are more promotion opportunities for those who actively participate in CS”.*
(Participant Q)


*“In our (mental rehabilitation) facility, we provide CS twice a month, and we write a CS diary for appraisal”.*
(Participant R)

The CS experience of participants led to continued or discontinued CS, depending on their “relationship with the clinical supervisor”. Those who accepted CS with a positive mind developed a better relationship with their clinical supervisors and gained further experience.


*“I learned hospitality from clinical supervisor A (during CS); overall mental health project including account, planning, and rehabilitation treatment of patients from clinical supervisor B; possibility of recovery of alcoholics from clinical supervisor C; and autonomy and professionalism from clinical supervisor D”.*
(Participant C)

### 3.5. Main Category: “Supporting Each Other and Becoming a Healthcare Expert”

The main category of CS experience of PMHNPs was “supporting each other and becoming a healthcare expert”. This process was divided into four stages: “asking for help”, “intensive training and accompanying”, “developing competency”, and “autonomous cooperation”. The participants asked for help in the difficult working environment of psychiatric nursing when they were new nurses. Like “caterpillars”, they hatched into “butterflies” to become competent PMHNPs with the help of intensive training by their clinical supervisors who were fully grown “butterflies”. Even after becoming butterflies, the participants ensured that they remained proficient healthcare professionals by being involved in support and collaboration, continuous CS, and learning, as shown in Figure 1.

#### Strategy: Process of “Supporting Each Other and Becoming a Healthcare Expert”

The main five interaction strategies used by the participants were “asking for help”, “intensive training and sharing with the clinical supervisor”, “modeling of the clinical supervisor and developing competencies”, “continuing self-reflection and learning”, and “participating in professional activities”. In this process, the ability to integrate theory and practice could grow depending on the strategy used; however, when situations became difficult, the adopted strategy was no longer applied or was applied intermittently, which was shown through varying levels of regression or turnover.

First, in “asking for help”, the participants experienced “difficulties working alone” in the poor working environment of psychiatric nursing and used the strategy of “asking for help” from their superiors or other experts. They felt afraid and nervous when the patients unexpectedly harmed themselves or others, and then required CS.


*“I followed my seniors during the rounds, listened to their questions on the mental health and sleep health statuses of patients, and did exactly the same”.*
(Participant M)

Second, in “intensive training and accompanying”, the participants had begun as general nurses and received intensive training for one year to become PMHNPs. At this stage, they mainly used the strategy of “intensive training and sharing together”. During their training, the participants were assigned clinical supervisors and showed differences in their professional development according to their clinical supervisors, personal characteristics, and institutional CS policies. Among these, being paired with a competent clinical supervisor had the greatest effect on their development.


*“I was clinically supervised for my attitude, voice, gaze, and actions every time, and I realized I was doing the same when I was clinically supervising the new supervisees”.*
(Participant E)


*“We wrote down the interview responses of the patients without missing any facial expressions or intonations and submitted them. A mental status examination was also conducted with the clinical supervisor to review whether the patient was adequately assessed”.*
(Participant B)

Third, in “developing competency”, which refers to Level 2 PMHNPs until they acquire Level 1 competency, the participants mainly used the strategies of “modeling and developing competencies” and “continuing self-reflection and learning” in various mental health settings. This aspect takes 5–10 years or more to develop depending on the type of strategy used by the participants; however, after 5 years of practical experience, they can apply for the PMHNP Level 1 certificate.

The participants who worked at community mental health facilities had difficulties adapting to the work environment, where the focus was not on treatment but on rehabilitation. Weekly multidisciplinary case meetings at mental health welfare centers provided an opportunity for a group CS time to review whether the services provided to patients were appropriate and to resolve difficult problems together. Most of the actual CS was peer-group conducted.

All but one of the participants who faced such limitations went to graduate schools to learn how to become more professional, which generally involved receiving individual expert training for autonomous counseling and rehabilitation.


*“When we were not sure which intervention was more appropriate, the clinical supervisors helped us review the reason for the conflicts and led us to reflect on the consistent principles and rules for the protection of our patients”.*
(Participant A)


*“At first, I thought two to three times and asked my clinical supervisor for feedback. Now, I think to myself more than 10 times and question “Are there any other ways?” and self-reflect; I also went to graduate school for further studies”.*
(Participant M)

Fourth, “autonomous cooperation” involved PMHNPs at Level 1 who had completed “developing competency”. During this period, the participants mainly used the strategies of “continuing self-reflection and learning” and “participating in professional activities”. The participants who had improved their competencies also further increased their capacity to integrate theory and practice to assess and observe potential risks. This led to autonomous cooperation with a high level of holistic nursing. The participants with expanded autonomy became competent clinical supervisors who could train case managers capable of caring for patients. Additionally, the participants actively participated in establishing a therapeutic environment by developing effective local networks and solidarity between multidisciplinary teams in order to lead changes in mental health policies. They helped in the occupational rehabilitation of well-recovered patients and provided greater support to help them recover interpersonal relationships that had been lost owing to their illness. They worked as proficient healthcare professionals supporting each other in hospitals and in local community mental nursing rehabilitation centers.


*“As a clinical supervisor, you have to provide objective and realistic information to the supervisees. To be a better clinical supervisor, I have to study and educate myself first before I supervise anyone else”.*
(Participant E)

In summary, the characteristics of the Korean PMHNP’s process of “supporting each other and becoming medical personnel” are shown in Table 3.

### 3.6. Consequences

The consequences of the interaction strategies of the participants were identified as “providing high quality nursing”, “maturing as an expert”, and “hoping to rebuild the CS system”.

In terms of “providing high-quality nursing”, the ultimate goal of CS was to strengthen the competency of PMHNPs by providing them with safe and high-quality nursing care education. The participants felt satisfied, as their patients believed in them, and good therapeutic relationships were maintained. In addition, they also felt rewarded in seeing their patients adapt successfully to the local community.


*“When you are seeing patients with addictions, you see the worst-case scenarios. As we manage these patients, we see ‘how beautiful it is to see development and growth through recovery,’ and this really made me feel rewarded. It kept me going and pushed me to work harder”.*
(Participant O)

In terms of “maturing as an expert”, this development involved more than 10 years of continued CS, learning, and observation in psychiatric nursing, through which the participants improved their professional knowledge, techniques, and capacity to integrate practice and theory to intuitively evaluate their patients. They developed creativity and autonomy to help treat their patients and their families through increased skill in offering targeted nursing interventions.


*“When I managed a patient (with schizophrenia) who had had the disease from a young age and lacked moral awareness, I could see the patient changing… I regularly evaluate the patients and send the results to the families and attending physicians to advocate for them”.*
(Participant F)

In terms of “hoping to rebuild the CS system”, the participants had experienced a frequent turnover but remained enthusiastic for continuous CS. They had had many positive experiences with CS and hoped that an effective CS system could be implemented.


*“We do not have education programs for supervision, and there is a lack of competent supervisors. This leads to incompetent PMHNPs and, in turn, hurts the patients”.*
(Participant B)


*“The CS system is unstable in nursing. In hospitals, we do not provide CS. We are just experiencing burn out. We should not be burnt out or exhausted. We need an effective supervision system”.*
(Participant R)

## 4. Discussion

This study used grounded theory to explore the CS experiences of PMHNPs in South Korea and determine the central phenomenon and its causes, the core category, subcategories, and strategies in relation to their CS experiences. The results showed that the central phenomenon was “lack of ability to integrate theory and practice”, and the core category was “supporting each other and becoming a healthcare expert”. The causes of the central phenomenon were “poor CS systems” and “difficulty working alone”.

The participants experienced “difficulty working alone” while nursing mentally ill people who committed suicide, self-harmed, or committed violence against others during work. This result supports the results of previous studies [26,27]. The participants in this study wanted to receive systematic supervision because of the “lack of an ability to integrate theory and practice”, that is, “a feeling of lack of competence”, “feeling lethargic”, “feeling burdened”, “feeling exhausted”, and “difficulty adapting to the diversity of mental health sites”. This finding corroborates the findings of previous studies [8,9,10,11,12,13,14,15] that emphasized the fact that PMHNPs’ CS participation is helpful for excellent clinical practice and continuous professional development.

The central phenomenon was addressed by using five strategies: “asking for help”, “intensive training and sharing with the clinical supervisor”, “modeling of the clinical supervisors and developing competencies”, “continuing self-reflection and learning”, and “participating in professional activities”. The active and repeated use of these strategies to address the central phenomena made it possible for the participants to “support each other to become healthcare professionals”, which was identified as the core category. This core category was divided into four subcategories, which tended to unfold in stages: “asking for help”, “intensive training and accompanying”, “developing competency”, and “autonomous cooperation”. However, this progression could be retarded if the PMHNPs stopped using these strategies or moved from hospital settings to community settings to start a new job.

Effective supervision is a key component of trainees’ professional development [8,9,10,16,19]. In this study, during the development of CS, the participants’ level of patient awareness and “self-other awareness” began with fear, changed to become more patient-centric, and expanded toward integration of a professional identity in the final stage. This is the result that supports the theory of the counselor’s integrated development model [28]. Therefore, specific, positive, and, above all, supportive CS should be provided in the starting phase in relation to “intensive training and accompanying” [8,9,10]. In addition, policies to support more relaxed CS in separate places during work hours need to be implemented first [9,18,29,30,31,32].

During the CS development process, the participants b egan with high motivational aspirations that fluctuated in the middle stage and became more stable and developed in the final stage. Their motivation was affected by specific intervening conditions that promoted or hindered CS: “personality characteristics”, “poor CS systems”, “institutional CS policy”, and “relationship with clinical supervisors”.

The “challenge-oriented type” participants were more active in CS, and their professionalism was found to improve because they had an honest openness, intellectual desire, enthusiasm, and compassion to help clients, which accords with previous findings [21,32]. The previous PMHNP curriculum in South Korea involved “intensive training and sharing with the supervisor” and was systematically managed. However, supervision is now superficial as in previous study [29] and, once their training courses are over, the PMHNPs no longer systematically receive CS. 

The participants became more dependent on the supervisors early in the relationship. As they progressed through their practice, the supervisees became more independent and autonomous and eventually reached the final “autonomous cooperation” stage. At Level 2 PMHNPs “developing competency” stage, the “challenge-oriented type” participants continued self-reflection, studied more in graduate school, and received CS in psychotherapy. They usually took 10–15 years to reach this “autonomous cooperation” stage.

The relationship with the clinical supervisor is a key factor for successful CS, and the quality of this relationship becomes more important over time [8,33]. In the current study, we found that meeting a competent clinical supervisor (relationship with clinical supervisors) was the most important factor. In general, the time spent with two to three competent clinical supervisors led to a peer-group-conducted CS relationship [34].

Concerns have been raised that the quality of treatment and service is poor due to a serious shortage of mental health experts worldwide [6]. In particular, South Korea has the highest suicide rate among Organisation for Economic Co-operation and Development countries, and it should be noted that senior PMHNPs are leaving the mental health field in response to the rapidly changing and more challenging environment that has emerged following the COVID-19 pandemic. To establish effective CS, appropriate CS policies need to be implemented and monitored [17,31], as also indicated by the findings of this study. 

In “Autonomous Collaboration”, the participants with an improved ability to integrate theory and practice were able to maintain good relationships with other professionals in multidisciplinary teams as mature professionals who provided high-quality nursing care to patients. These findings support those of previous studies that found that CS helped ensure high-quality service to patients and supported professional growth and development [8,35]. Although the participants with enhanced competency actively participated in professional activities as clinical supervisors, they identified that a lack of CS awareness and the uncertain theoretical status of CS concepts are obstacles to CS development [8,10,35]. In South Korea, it was found that the professionalism was improved by actively developing the CS with the passion and effort of each participant. It was also argued that the government and various medical institutions that actively invest in and support high-quality medical care through professional enhancement through CS are the most important.

In Europe (e.g., the UK) and Australia, the importance of CS is recognized, systematic guidelines are provided, and CS performance monitoring and CS effect studies are increasing [12,15,18,31,36,37]. The participants in this study hope to establish systematic CS, such as providing CS guidelines and education, as in Europe or Australia. In addition, it was confirmed that there were many cases of PMHNPs leaving the company after failing to adapt to various administrative tasks of community mental rehabilitation facilities. This differs from previous research [36] that community mental health nurses are more likely to receive CS than mental health nurses in clinical settings. In Korea, due to the general lack of awareness of psychiatric nurses, professional senior PMHNPs disappear from the community due to frequent the turnover or the transfer of mental health nurses, and the multidisciplinary approach to case management of severely mentally ill patients in the community is causing serious problems. PMHNPs in South Korea hopes to establish a community CS education system as soon as possible.

The significance of this study is that it is the first grounded theory study to explore the development process of “supporting each other and becoming a healthy expert” by using five strategies to solve the central phenomenon of a “lack of ability to integrate theory and practice”. This study will be the starting point for the development of mental health nursing practice in South Korea. However, additional in-depth research on turnover will be necessary at the ”developing competency” stage.

## 5. Limitations

This study is the first one to analyze the process of CS, as experienced by PMHNPs, using grounded theory in South Korea. However, it also has certain limitations. First, this study collected data from South Korea, where CS among PMHNPs is poor. Second, the study results were analyzed based on the retrospective statements of senior PMHNPs, who received CS from three supervisors, on average, for more than 10 years. Third, the supervisors’ points of view may have been overlooked because the CS process was analyzed from the supervisees’ perspective. Fourth, the fact that the study was conducted before COVID-19 means that more data are expected to be collected on the supervision of mental nursing in a pandemic situation. Fifth, this study recruited subjects by the snowball method, and the participants were mainly female, but it seems necessary to consider the gender ratio in the future.

Therefore, bias may have affected the results, making it difficult to generalize. To overcome this limitation, various follow-up studies, which could focus on the CS experience of beginners or supervisors, supervisor training program development, and tool development for the evaluation of CS effectiveness, are needed.

## 6. Conclusions

In this study, the CS experience of South Korean PMHNPs was comprehensively analyzed by using grounded theory. The central phenomenon identified was the “lack of ability to integrate theory and practice”, and the core category derived was “supporting each other and becoming a healthcare expert”. The central phenomenon was found to be caused by “difficulty working alone” and “poor CS systems”. Intervening conditions such as “personality characteristics”, “institutional CS policies”, and “relationship with clinical supervisors” affected the strategies adopted in relation to CS, which acted either to advance or inhibit CS development. The process of “supporting each other and becoming a healthcare expert” was divided into four subcategories, which tended to unfold in stages: “asking for help”, “intensive training and accompanying”, “developing competency”, and “autonomous cooperation”. In terms of these findings, it is clear that, although the professionalism of the PMHNPs was improved through their own efforts, an innovative systematic CS policy is required to ensure a more effective mental health nursing service.

## Figures and Tables

**Figure 1 ijerph-19-15904-f001:**
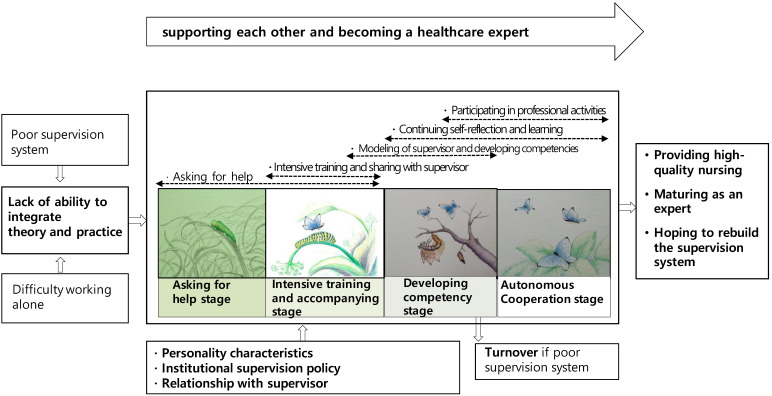
A model of experiences of psychiatric mental health nurse practitioners in clinical supervision.

**Table 1 ijerph-19-15904-t001:** General characteristics of the study participants (*N* = 19).

Classifications	Work Experience			Experience in Supervision (Years)	Certificates	Education Level
	Place of Work *		Experience in Mental Health Nursing		PMHNP (Level)	
	Hospital	Local Community				
A	○	○	30	20 years+	Level 1	Doctorate
B	○	×	17	10 years+	Level 1	Doctorate
C	○	○	22	15 years+	Level 1	Master’s
D	○	×	5	1 years	Level 2	Bachelor’s
E	○	○	31	20 years+	Level 1	Master’s
F	○	○	41	15 years+	Level 1	Master’s
G	○	×	13	5 years+	Level 1	Master’s
H	○	○	17	10 years+	Level 1	Doctorate
I	○	○	22	10 years+	Level 1	Doctorate
J	○	○	9	3 years	Level 2	Master’s
K	○	○	8	1 year	Level 1	Master’s
L	○	○	22	4 years	Level 1	Master’s
M	○	○	22	20 years+	Level 1	Master’s
N	○	○	7	3 years	Level 2	Master’s
O	○	○	21	15 years+	Level 1	Doctorate
P	○	○	17	5 years+	Level 2	Master’s
Q	○	○	40	20 years+	Level 1	Master’s
R	○	○	19	10 years+	Level 1	Doctorate
S	○	○	14	10 years+	Level 1	Master’s

* Workplace: Yes (○); No (×). Note: PMHNP, psychiatric mental health nurse practitioner.

**Table 2 ijerph-19-15904-t002:** Paradigm, categories, and subcategories of experiences in clinical supervision of psychiatric mental health nurse practitioners.

Paradigm	Categories	Subcategories (Frequency)
Causal condition	Difficulty working alone	Poor working conditions (51)
		Being fearful and tense (25)
Central phenomenon	Lack of ability to integrate theory and practice	Feeling of lack of competence (193)
		Feeling lethargic (21)
		Feeling burdened (103)
		Feeling exhausted (33)
		Difficulty adapting to the diversity of mental health sites (52)
Contextual condition	Poor supervision system	Superficial supervision (77)
		Hoping for practical systematic supervision (319)
Intervening condition	Personality characteristics	Dedicated to development (210)
		Enduring with a positive mind (106)
	Institutional supervision policy	Feeling motivated (45)
		Feeling demoralized (26)
	Relationship with supervisor	Investigating and coordinating with each other (57)
		Learning from negative lessons (88)
Action and interaction strategy	Asking for help	Relying on superiors (26)
		Learning rehabilitation techniques (86)
	Intensive training and sharing with supervisor	Intensively training together (334)
		Getting professional training (340)
		Supported and encouraged by the supervisor (154)
		Receiving evaluation and feedback from supervisors (95)
		Training in various clinical skills (234)
		Training in reflection skills (87)
	Modeling of supervisor and developing competencies	Developing competency with expert supervision (94)
		Modeling the supervisor (37)
	Continuing self-reflection and learning	Continuing self-reflection (22)
		Continued learning in a variety of ways (29)
	Participating in professional activities	Acting as a supervisor (62)
		Continued research and academic exchanges (64)
Consequence	Providing high-quality nursing	Being rewarded (66)
		Being satisfied (122)
	Maturing as an expert	Integrating theory and practice (197)
		Having satisfactory professional relations (27)
	Hoping to rebuild the supervision system	Hoping for a competent supervisor (293)
		Wanting a systematic supervision system (285)

**Table 3 ijerph-19-15904-t003:** Characteristics of the process of “becoming health care experts by supporting each other”.

Stage	Asking for Help	Intensive Training and Accompanying	Developing Competency	Autonomous Cooperation
Certificate	Registered Nurse	Trainee for PMHNP	PMHNP level 2	PMHNP level 1
Period	1 or 2 years	1 year	5 years or more	10 years or more
Legal education and supervision (hours/year)	8 h none	150 h 850 h	12 h or more none	12 h or more none
Education institution	Basic knowledge (workplace)	General knowledge (training institution)	Optional knowledge (personal/graduate school, etc.)	Optional knowledge (autonomous)
Action and interaction strategy	Asking for help	Intensive training and sharing with supervisor Modeling of supervisor and developing competencies	Modeling of supervisor and developing competencies Continuing self-reflection and learning Participating in professional activities	Continuing self-reflection and learning Participating in professional activities

## Data Availability

Not applicable.

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
