# Peer review of "The Experiences of Psychiatric Mental Health Nurse Practitioners with Clinical Supervision in South Korea: A Grounded Theory Approach"

_ijerph, 2022, doi:10.3390/ijerph192315904_

Round 1

Reviewer 1 Report

Dear Editor,

Thank you for the opportunity to review the manuscript titled: The experiences of psychiatric mental health nurse practitioners with clinical supervision in South Korea: A grounded theory approach. Please find my feedback detailed below.

My main concern with the manuscript is that the information is outdated. The data was collected in January – April 2018 and would no longer be contextually relevant or applicable, particularly in the context of the COVID-19 pandemic. The authors also state that mental health care nurses “do not receive clinical supervision even in crisis situations”. This suggests that the stressors they would encounter may be even more pronounced during a pandemic.

Introduction

The introduction needs expansion. The authors are using a grounded theory approach and it would be necessary to cite and discuss existing theoretical models that underpin their research topic (e.g., clinical supervision models) and the gaps that their research aims to fill.

Existing qualitative studies describing the experiences of psychiatric mental health practitioners need to be reviewed.

Methodology

All of the research participants are women. The authors need to clarify if this is due to a skewed gender ratio of practitioners in the country. It also needs to feature under the limitations section.

Level of experience: Three of the participants have level 2 certificates while 16 have level 1 certificates. Only one participant has a Bachelor’s degree while all other participants have either a Master’s or doctoral degree. This indicates key differences in their level of training which would impact on their supervision needs and experiences of clinical supervision. This needs to be critically discussed given that in qualitative research, it is more rigorous to have a homogenous sample.  

The work setting and the types of cases (e.g., common mental health disorders or more severe psychopathology) seen by these practitioners needs to be discussed in more detail as this would influence their supervision related needs. It also has a bearing on their experiences within the work setting.

Results and Discussion

These sections do not necessarily lend anything new to the existing literature in this area. The authors have discussed the main themes arising from the participants accounts but there is a lack of depth and nuance. Figure 1 demonstrates a caterpillar transforming into a butterfly and is an overly simplistic way of characterising the experiences of participants and is not grounded in theory. It is unclear how this model was arrived at and what it aims to contribute.

The purpose of using a grounded theory approach is either to develop a new theory or to add in a substantive way, to an existing theory. Regrettably this has not been achieved.

Reviewer 2 Report

This paper presents a qualitative study of experiences surrounding clinical supervision (CS) among psychiatric mental health nurse practitioners (PMHNPs) in South Korea. Strengths of this study include a well-articulated theoretical background and a clear set of implications for practice. I have a few comments that I think could be addressed to strengthen the paper further.

1. If appropriate, it would be informative to provide a frequency table for the themes or groups of themes that arose in various categories.

2. Since there is little previous literature in this area in the South Korean context, it may be important to reflect briefly on whether or not there are any cultural differences between the Korean context and other regions that may have influenced these outcomes.

3. Please include a paragraph addressing limitations in the present research.

Reviewer 3 Report

This is a very comprehensive article about a topic that is not getting much interest scientifically nor in clinical practise but important for the sake of improving our service to the patients and  to improve working conditions for the staff.The data collection and analyses  methods are well described.The is an amount of illustrative citations to underline the  findings.An interesting moment is the final validation of the results and interpretations by a number of experienced researchers and   nurses.

 My only  minor comment is that the  participants are all  very well educated and well trained with a long professional experience.It would have been interesting also to include younger nurses in the beginning of their career but this  might be another study.

Round 2

Reviewer 1 Report

Dear Editor, 

My concerns about the methodology of the paper remain. The data remains outdated and the paper lacks originality and merit.

Author Response

to. Reviewer 1

Thank you sincerely for your comments. Please understand that the data are outdated and incomplete. We have already learned through research that many countries, including your own , have established the Supervision system and are conducting research. However, in Korea, this research is in the initial stage, so I think it is a task that needs to be continued in the future. Although the research is considerably insufficient, we expect to be able to provide basic guidelines necessary for the domestic clinical field based on the research.

Thanks again for your comment, It is difficult to completely complement, but we have added the following to the Discussion of 12p. Thank you.

The significance of this study is that it is the first grounded theory study to explore the development process of 'supporting each other and becoming a healthy expert' by using five strategies to solve the central phenomenon of a 'lack of ability to integrate theory and practice. This study will be the starting point for the development of mental health nursing practice in South Korea. However, additional in-depth research on turnover will be necessary at the 'developing competency' stage.
